# Predictors of stent dysfunction in patients with bilateral metal stents for malignant hilar obstruction

**Hoonsub So[1], Chi Hyuk Oh [2], Tae Jun Song[3]\*, Sung Woo Ko[3], Jun Seong Hwang[4], Dongwook Oh[3], Do Hyun Park[3], Sang Soo Lee[3], Dong-Wan Seo[3], Seok Ho Dong[2], Sung Koo Lee[3], Myung-Hwan Kim[3]**

**1** Department of Internal Medicine, Ulsan University Hospital, University of Ulsan College of Medicine, Ulsan, Republic of Korea, **2** Division of Gastroenterology and Hepatology, Department of Internal Medicine, Kyung Hee University Hospital, Seoul, Republic of Korea, **3** Division of Gastroenterology, Department of Internal Medicine, University of Ulsan College of Medicine, Asan Medical Center, Seoul, Republic of Korea, **4** Division of Gastroenterology, Department of Internal Medicine, Inje University College of Medicine, Haeundae Paik Hospital, Busan, Republic of Korea

☯ These authors contributed equally to this work.
\* drsong@amc.seoul.kr

**Data Availability Statement:** All relevant data are within the manuscript and its Supporting Information files.

## Abstract

### Introduction

For unresectable hilar obstruction, restoring and maintaining biliary ductal patency are crucial for improved survival and quality of life. The endoscopic placement of stents is now a mainstay of its treatment, and bilateral stenting is effective for biliary decompression. This study aimed to determine the clinical outcomes of bilateral metal stent placement using large cell-type stents and the clinical predictors of stent dysfunction in patients with malignant hilar obstruction.

### Methods

We performed a retrospective analysis of patients who underwent bilateral metal stent placement using two large cell-type stents at two academic teaching hospitals between September 2017 and February 2019. The primary outcome was stent dysfunction. Secondary outcomes included predictors related to stent dysfunction and overall survival.

### Results

The study included 87 patients who underwent bilateral metal stent placement for malignant hilar obstruction. Technical success and clinical success were achieved in 80 patients (92.0%) and 83 patients (95.4%), respectively. During the follow-up period (median: 201, range: 18–671 days), stent dysfunction occurred in 42 patients (48.3%), and the median stent patency was 199 days (95% confidence interval [CI]: 181–262). In univariate analysis, age, cholangitis before stent insertion, and subsequent chemotherapy were found to be associated with the cumulative risk of stent dysfunction. In multivariate analysis, cholangitis before stent insertion (hazards ratio [HR]: 2.26, 95% CI: 1.216–4.209, P = 0.010) and

**Funding:** The authors received no specific funding for this work.

**Competing interests:** The authors have declared that no competing interests exist.

subsequent chemotherapy (HR: 0.250, 95% CI: 0.130–0.482, P<0.001) remained as statically significant factors associated with the cumulative risk of stent dysfunction. The median overall survival was 288 days (95% CI: 230–327).

## Conclusion

The bilateral placement of large cell-type stents for malignant hilar obstruction was effective with high technical and clinical success rates and acceptable patency. Cholangitis before stent insertion was associated with shorter patency, and subsequent chemotherapy was associated with longer stent patency.

## Introduction

The perihilar type is the most common type of cholangiocarcinoma (CCA), representing 50% of CCA cases [1]. Perihilar biliary obstruction can also be caused by gallbladder carcinoma, hepatocellular carcinoma, and many other metastatic diseases. In patients with unresectable malignant hilar biliary obstruction, restoring and maintaining biliary ductal patency is the key to improved survival and quality of life [2]. The endoscopic placement of stents is currently a mainstay of its treatment [3, 4]. Improvements in chemotherapy, radiotherapy, and local ablative therapy have led to longer survival, suggesting that longer stent patency is an important factor in these palliative therapies.

For hilar malignant strictures, especially for Bismuth types II–IV strictures, the European Society of Gastrointestinal Endoscopy (ESGE) indicates that draining more than 50% of the liver volume, which requires a self-expandable metal stent (SEMS), is important for effective drainage and longer median survival [5]. Several studies have reported that bilateral stenting with a SEMS is a good way to maximize the drainage volume of the liver, leading to more durable and effective drainage. Bilateral SEMS placement can be performed using a stent-in-stent (SIS) or stent-by-stent (SBS) method. Both methods are challenging as the space in the common bile duct (CBD) is limited, and the mesh of the stent may not allow for a second stent deployment. Recently, manufacturers have developed uncovered metal stents with a large cell, which allow endoscopists to more easily perform bilateral SEMS deployment [6]. However, in contrast to fully covered metal stents, uncovered stents are more likely to be re-obstructed by tumor ingrowth. Several risk factors have been reported for the recurrent obstruction of metal stents; however, the types and methods of stent deployment vary. The present study aimed to access the patency of large cell-type stents and investigate the factors associated with stent obstruction after bilateral SEMS placement for malignant hilar obstruction.

## Materials and methods

### Patients

Data from patients who underwent bilateral SEMS insertion using a Niti-S® large cell D-type stent (LCD; Taewoong Medical, Gimpo, Korea) between September 2017 and February 2019 at Asan Medical Center and Kyung Hee University Hospital were reviewed retrospectively. This was a retrospective study that was conducted in accordance with the ethical guidelines of the 1975 Declaration of Helsinki and approved by the institutional review board of Asan Medical Center (2019–1349) and Kyung Hee University Hospital (2019-08-054). Given its retrospective nature, written informed consent to access the clinical data was not required by the

board. All data were fully anonymized for collection and analysis, and all included populations were Korean.

## Procedure

Before the procedure, resectability was evaluated with a multidisciplinary approach by the surgeon and oncologist. Subsequently, the endoscopist chose candidates for bilateral stenting considering the tumor extent, intrahepatic duct dilatation, and liver volume. If the patient was admitted with cholangitis, nasobiliary drainage was performed first, and replacement with bilateral metal stents was performed after improvements in their cholangitis and jaundice. All procedures were performed by three expert endoscopists (S.T.J, O.D.W., and O.C.H) who perform more than 1,000 endoscopic retrograde cholangiopancreatography (ERCP) procedures a year.

After sedating the patients with midazolam and meperidine, a duodenoscope (TJF-260V; Olympus Medical Systems Corporation, Tokyo, Japan) was used for the procedures under fluoroscopic guidance. After biliary cannulation, two 0.025-inch guidewires were placed in the left and right hepatic duct. If the left duct was missing, wires were placed in the right anterior and right posterior segmental duct (RASD and RPSD) at the same time. All procedures used a Niti-S® large cell D-type stent (Taewoong Medical, Gimpo, Korea), and the length (6 cm, 8 cm, or 10 cm) and diameter (8 mm or 10 mm) of the SEMSs were chosen at the endoscopist's discretion after evaluating the stricture of each bile duct. After deploying the first SEMS in the left hepatic duct, the guidewire was not retrieved; then, it followed the tract of the previously placed wire passing through the mesh (Fig 1). In this approach, SIS placement was achieved with the formation of a Y-shape. If the SIS method failed, the SBS method was employed for rescue. All procedures were completed in a single session.

## Outcomes and definitions

The primary outcome was stent dysfunction. Stent dysfunction was defined as obstruction by tumor ingrowth, hemobilia, or a sludge/stone. If there was any new obstruction away from the stent (e.g., distal CBD), it was not considered as stent dysfunction. Stent patency was calculated from the day of insertion to stent dysfunction. If the patient died before stent dysfunction, they were censored. The secondary outcomes were technical success, clinical success, adverse events, risk factors related to stent dysfunction, and overall survival. Technical success was defined as the satisfactory deployment of bilateral stents using the SIS method. Switching to the SBS method was considered as a technical failure as it is regarded as a rescue method. Clinical success was defined as a decrease in the bilirubin level to a normal level or less than a quarter of the pretreatment level within the first month [7]. The maximum levels of alkaline phosphatase (ALP) and bilirubin during admission and the procedure were recorded. The day of diagnosis was defined as the day when the biopsy result was reported. In cases of metastatic cancers, the diagnosis day was designated as the day of diagnosis of related symptoms (fever, jaundice) or the first biliary drainage procedure. The presence of cholangitis before stent insertion was recorded. Cholangitis was defined as a fever with signs of jaundice with/without right upper quadrant pain. Adverse events were classified as either intra-procedure or post-procedure (up to 14 days according to a lexicon for endoscopic adverse events) [8]. Overall survival was calculated from the day of the procedure to the day of death or the last day of follow-up.

## Statistical analysis

Descriptive statistics including the median, range, and percentage were calculated. We estimated the cumulative risk of stent dysfunction and overall survival using the Kaplan-Meier

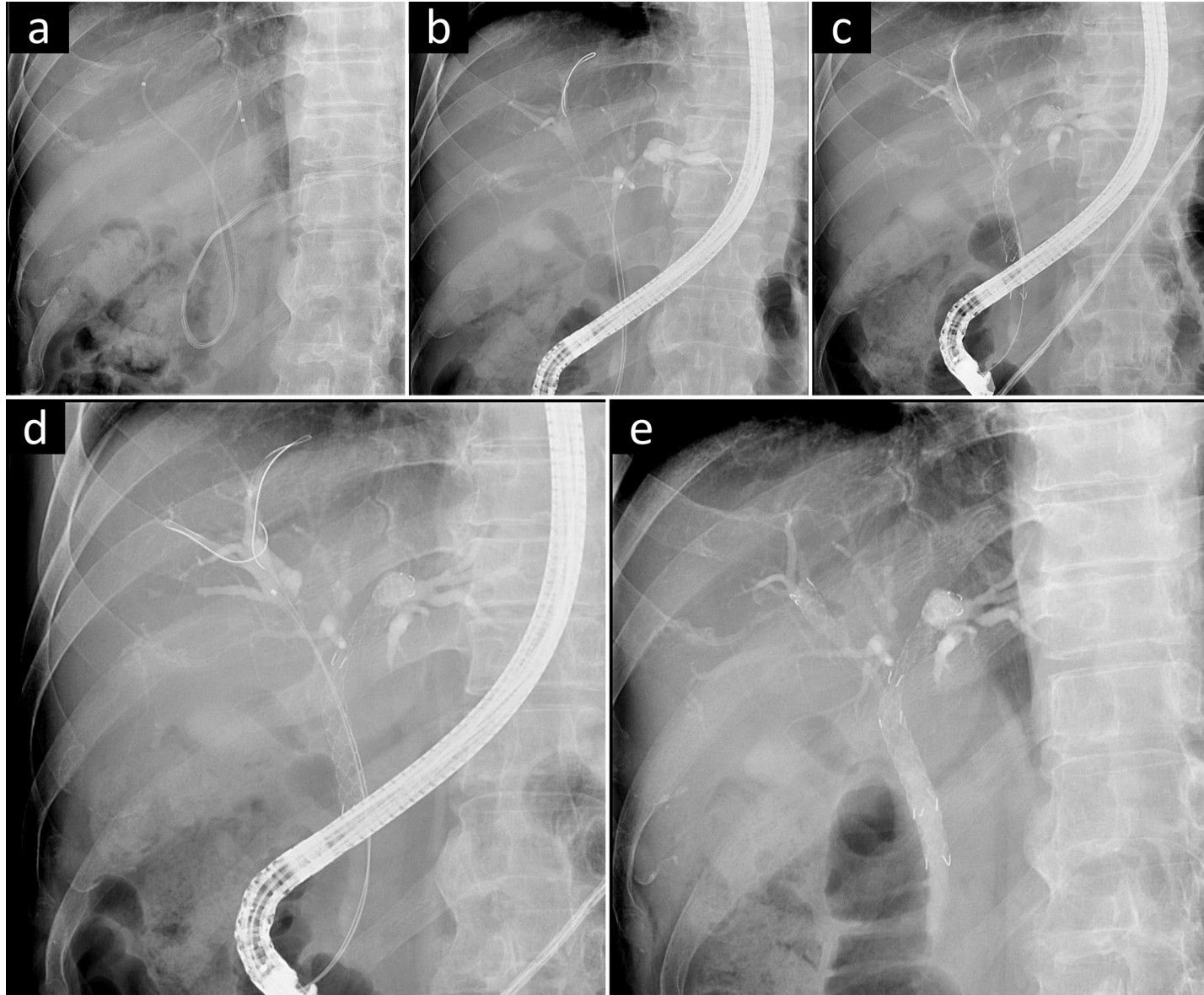

**Fig 1. A case of bilateral metal stenting for malignant hilar obstruction.** a. Fluoroscopic view with bilateral nasobiliary drainage; b. After biliary cannulation, two 0.025-inch guidewires were placed in the left and right hepatic duct; c. The first stent was deployed in the left duct, and the remaining guidewire was inserted through the cell of the left stent with the guidewire in the right duct as the reference; d. A catheter was introduced to confirm resistance and to evaluate the right duct again; e. The second stent was then deployed as the stent-in-stent method.

method. To identify associations between stent dysfunction and possible risk factors, we fitted both univariate and multivariate Cox proportional hazards regression models. In the multivariate regression model, the final model was selected by stepwise regression. Likelihood ratio tests were performed to test the significance of the categorical variables. The proportional hazards assumption in the Cox regression model was examined by the Schoenfeld residuals test, and the assumption was deemed satisfied in all final models (P>0.05). All reported P-values were two-sided, and a P-value of less than 0.05 was considered to indicate statistical significance. Data were analyzed using R program Ver. 3.5.3. (R Foundation for Statistical Computing, Vienna, Austria; http://www.R-project.org).

## Results

### Patients

A total of 112 patients underwent bilateral SEMS insertion for malignant hilar obstruction, and 25 patients were excluded because they were transferred to other hospitals for supportive care after the procedure. Finally, 87 patients were included in the analysis (Asan Medical Center: 73, Kyung Hee University Hospital: 14). All patients except for 2 patients were diagnosed with a biopsy, and 2 patients were diagnosed as having CCA based on imaging and the clinical course after the failure of pathologic confirmation. Their median age was 67 years (range: 47–87), and the male-to-female ratio was 49:38. Overall, 69, 9, and 9 patients were diagnosed as having CCA, gallbladder cancer, and metastatic cancers, respectively. The types of metastatic cancers were colorectal cancer (n = 4), pancreatic cancer (n = 1), NET (n = 1), ovarian cancer (n = 1), sarcomatoid carcinoma (n = 1), gastric cancer (n = 1), and bladder cancer (n = 1). Bismuth classifications were as follows: type II (n = 12, 13.8%), type III (n = 26, 29.9%), and type IV (n = 49, 56.3%). A total of 40 patients (46.0%) had a history of hilar stenting. The mean time from diagnosis to bilateral metal stenting was 43 days (SD: 65.5). A total of 36 patients (41.4%) had cholangitis before stent insertion. The levels of ALP and bilirubin before the procedure were 427 IU/dL (63–1757) and 3.4 mg/dL (0.3–26.6), respectively. A total of 49 patients underwent chemotherapy after the procedure. The patient characteristics are summarized in Table 1.

**Table 1. Baseline characteristics of patients.**

| Patient-related characteristics | Value (n = 87) |
|---|---|
| Age, y | 67 (47–87) |
| Male:Female | 49:38 |
| Diagnosis | |
| Cholangiocarcinoma (CCA), n | 69 |
| Gallbladder cancer | 9 |
| Others[¥] | 9 |
| Bismuth type[*] | |
| Type II | 12 |
| Type III | 26 |
| Type IV | 49 |
| Previous history of hilar stenting | 40 (46.0) |
| Time from diagnosis to stent insertion, days | 43 (65.5) |
| Procedure-related characteristics | |
| Cholangitis before stent insertion | 36 (41.4) |
| Laboratory examination | |
| Alkaline phosphatase | 409 (63–1757) |
| Bilirubin | 4.6 (0.3–26.6) |
| Subsequent therapy | |
| Chemotherapy | 49 (56.3) |
| Radiotherapy | 17 (19.5) |
| Best supportive care | 27 (31.0) |

All values are presented as the median (range), mean (standard deviation), and number (%).

[a] Bismuth type was classified according to the Bismuth-Corlette classification of perihilar cholangiocarcinoma.

[b] Others include colorectal cancer (n = 4), pancreatic cancer (n = 1), NET (n = 1), ovarian cancer (n = 1), sarcomatoid carcinoma (n = 1), gastric cancer (n = 1), and bladder cancer (n = 1).

### Clinical outcomes and adverse events

Technical success and clinical success were achieved in 80 patients (92.0%) and 83 patients (95.4%), respectively. A total of 7 patients with technical failure were rescued by the SBS technique. A total of 4 patients had clinical failure. They were managed with supportive care (n = 1), additional endoscopic retrograde biliary drainage (ERBD) stent insertion (n = 1), PTBD (n = 1), or EUS-biliary drainage (n = 1). A total of 7 patients (9.6%) had post-procedure adverse events, including cholangitis (n = 5), stent expansion failure (n = 1), and cholecystitis (n = 1). Stent dysfunction occurred in 42 patients (48.3%) during the follow-up period (median: 201, range: 18–671 days), and the median stent patency was 199 days (95% CI: 181–262). The etiologies of stent dysfunction were as follows: tumor ingrowth (n = 36), sludge/stone (n = 2), hemobilia (n = 3), and insufficient expansion (n = 1). Among these patients, 41 patients were managed successfully by endoscopic procedures; however, 1 patient died after percutaneous drainage due to disease progression. Death occurred in 51 patients (58.5%) during follow-up. The median overall survival was 288 days (95% CI: 230–327). The results are summarized in Table 2.

### Factors associated with SEMS dysfunction

In univariate analysis, the presence of cholangitis before stent insertion and subsequent chemotherapy were found to be associated with stent dysfunction. In multivariate analysis, both remained as significant factors associated with stent dysfunction (presence of cholangitis before stent insertion, HR: 2.26, 95% CI: 1.216–4.209, P = 0.010; subsequent chemotherapy, HR: 0.25, 95% CI: 0.130–0.482, P<0.001) (Fig 2). The results are summarized in Table 3.

## Discussion

In perihilar CCA and metastatic hilar diseases, the majority of patients are presented with an unresectable status, and decompression of the extrahepatic biliary obstruction is recommended via ERCP rather than by a percutaneous approach or surgery [9]. This is known to be a beneficial palliative treatment compared with surgery in terms of survival and cost [10]. Both plastic stents and SEMSs are commonly used for biliary drainage. Plastic stents are relatively cheap and can easily be exchanged. However, plastic stents have a high rate of obstruction due

**Table 2. Outcomes and adverse events of bilateral LCD stent insertion for malignant hilar obstruction.**

| Outcomes | n = 87 |
|---|---|
| Technical success, n (%) | 80 (92.0) |
| Clinical success, n (%) | 83 (95.4) |
| Stent dysfunction, n (%) | 42 (48.3) |
| Median stent patency, days | 199 (95% CI: 181–262) |
| Death, n (%) | 51 (58.6) |
| Median overall survival, days | 288 (95% CI: 230–327) |
| Adverse events | |
| Intra-procedure | None |
| Post-procedure, n (%) | 7 (8.0) |
| Cholangitis | 5 (5.8) |
| Stent expansion failure | 1 (1.1) |
| Cholecystitis | 1 (1.1) |

CI, confidence interval.

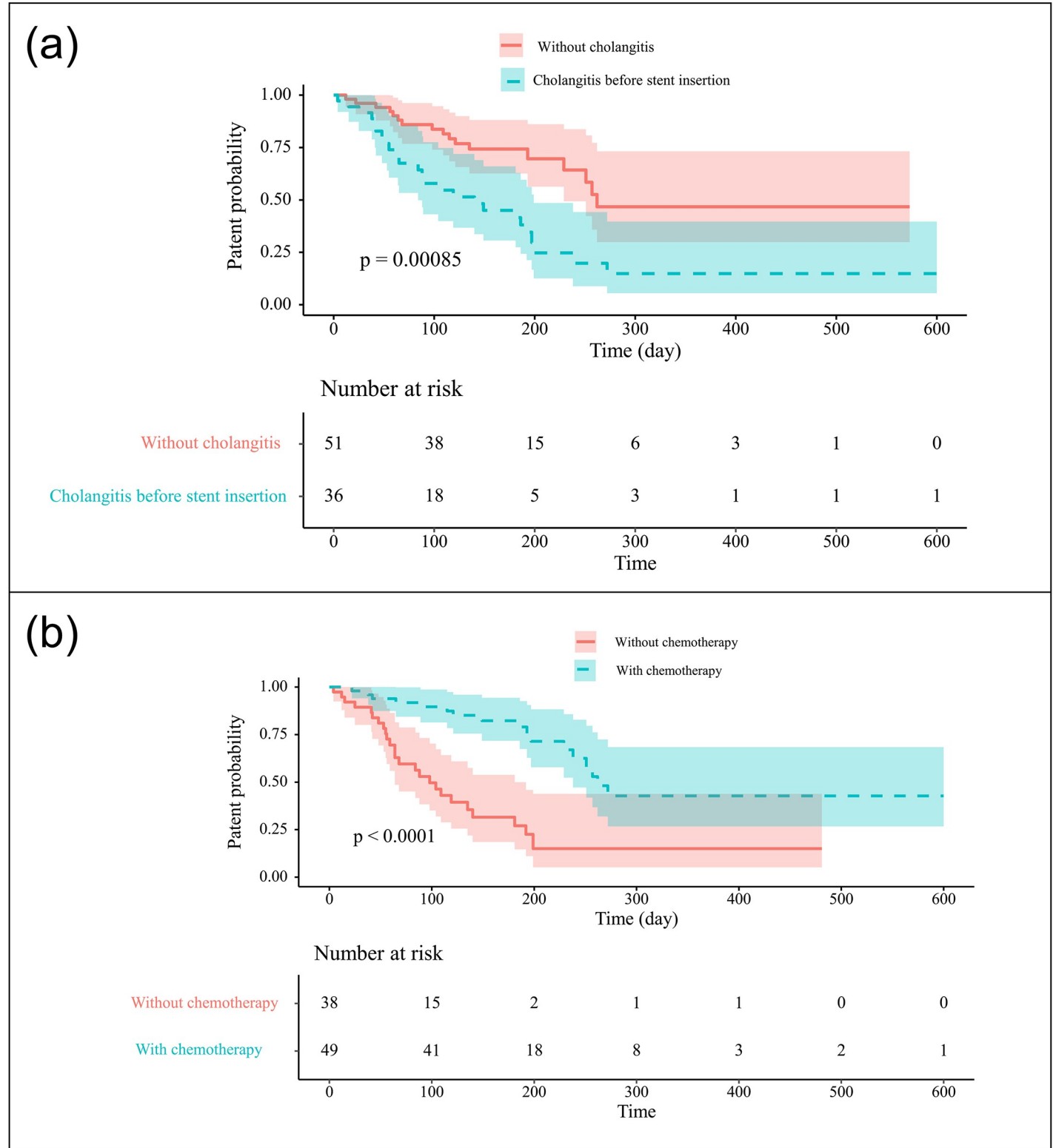

**Fig 2. Kaplan-Meier curve showing the cumulative stent patency.** (a) Presence of cholangitis before the procedure; (b) Subsequent chemotherapy.

**Table 3. Univariate and multivariate analysis of risk factors of stent dysfunction.**

| Variables | Univariate | | | Multivariate | | |
|---|---|---|---|---|---|---|
| | HR | LCI–UCI | P | HR | LCI–UCI | P |
| Sex | 1.70 | 0.894–3.220 | 0.106 | | | |
| Age | 1.03 | 0.998–1.057 | 0.070 | | | |
| ALP | 1.00 | 0.999–1.001 | 0.872 | | | |
| Bilirubin | 1.02 | 0.974–1.069 | 0.388 | | | |
| Diagnosis | | | | | | |
| CCA | reference | | | | | |
| GB | 1.18 | 0.417–3.311 | 0.760 | | | |
| Others | 0.65 | 0.199–2.112 | 0.472 | | | |
| Days from diagnosis to stenting | 1.00 | 0.999–1.007 | 0.194 | | | |
| Cholangitis before stent insertion | 2.75 | 1.480–5.096 | 0.001 | 2.41 | 1.282–4.528 | 0.006 |
| Previous stent placement | 1.80 | 0.975–3.310 | 0.061 | | | |
| Chemotherapy | 0.23 | 0.123–0.443 | <0.001 | 0.26 | 0.135–0.510 | <0.001 |
| Radiotherapy | 0.57 | 0.261–1.216 | 0.142 | | | |

ALP, alkaline phosphatase; CCA, cholangiocarcinoma; GB, gallbladder

to their small diameter and thus must be replaced frequently. On the other hand, SEMSs are superior to plastic stents in terms of stent patency. In our retrospective analysis, SEMSs showed better patency compared with the patency of plastic stents (HR 2.54; 95% CI: 1.600–4.026, P<0.0001) (S1 File). Therefore, SEMSs are recommended for palliative drainage; especially for malignant hilar obstruction, uncovered SEMSs are preferred because covered stents may induce cholecystitis by obstructing the cystic duct orifice. Uncovered stents are less likely to migrate; however, they are vulnerable to tumor ingrowth. Moreover, in contrast to fully covered metal stents, they are difficult to remove. Therefore, before deploying a bilateral uncovered SEMS, clinicians should determine the optimal timing of stent placement, procedure indication, and stent type. It is important to identify the predictive factors that may affect stent patency after bilateral metal stent placement. In our study, the presence of cholangitis before stent insertion (HR: 2.41, 95% CI: 1.282–4.528, P = 0.006) was associated with shorter patency, and subsequent chemotherapy (HR: 0.263, 95% CI: 0.135–0.510, P<0.001) was associated with longer patency. Laboratory examinations did not show any association with stent patency.

Several other studies have also reported risk factors related to metal stent dysfunction in cases of malignant perihilar obstruction. Miura et al. identified gallbladder carcinoma and left-sided SEMSs as risk factors. Moreover, cholangitis before SEMS insertion was a risk factor for recurrent biliary obstruction (HR: 11.44, 95% CI: 4.48–32.35, P<0.001) [11]. Their study design was different from ours as they included patients with unilateral insertion. However, stent patency appeared to be affected in patients with cholangitis compared with patients without cholangitis.

Cholangitis could produce large amounts of inflammatory materials resulting from obstruction by sludge and bile stasis. Furthermore, bacterial colonization could lead to bacterial biofilm formation, which triggers stent occlusion [12]. The resolution of cholangitis by drainage with sufficient antibiotic administration and nasobiliary drainage would be important before inserting bilateral metal stents.

In a retrospective study by Naitoh et al. comparing the SBS and SIS methods, they found that anti-tumor therapy was a factor associated with longer stent patency (HR: 0.31, 95% CI: 0.11–0.91, P = 0.034) [13]. Our study also showed better patency in patients undergoing

chemotherapy. In Heo's study, Bismuth type IV (vs. II and III), immediate complications within 72 h of stent placement, and baseline bilirubin (>6.1 mg/dL) were associated risk factors for stent occlusion. A baseline bilirubin higher than 6.1 mg/dL remained significant after multivariate analysis (HR: 2.606, 95% CI: 1.223–5.553, P = 0.013) [14]. The discrepancy with our results can be explained by the heterogeneous diseases, small number of patients, and methods used for analysis and data collection.

In perihilar CCA, 90% of patients are presented with painless jaundice, and 10% of patients are presented with acute cholangitis [15]. Overall, bilateral stents should be inserted after cholangitis treatment, and prompt chemotherapy should be initiated for longer stent patency. The necessity of bilateral stenting is another conflicting issue; however, many datasets favor bilateral stenting [16–18]. The advantage of bilateral stenting is that it is more physiological, and it can drain the maximal liver volume. A recent randomized trial showed a higher clinical success rate (84.9 vs. 95.3%, P = 0.047) and a lower reintervention rate (60.3 vs. 42.6%, P = 0.049) without any additional adverse events [17]. Naitoh et al. reported that patients with Bismuth type IV, IIIa, and II needed multiple biliary drainage before undergoing surgery; thus, bilateral stenting may reduce the number of drainage procedures in patients with these Bismuth types.

The drawbacks of bilateral stenting, especially with the SIS method, are its technical difficulty, longer procedure time, and higher cost. However, the stent used in our study (Niti-S large cell D-type stent) showed a technical success rate of 92.0% with the SIS method and a complication rate of 8.0%. All complications were managed successfully. The unique chain-like connection of each stent cell contributes to a low axial force, and its uniformly large cell size (7 mm) allows the guidewire and the second stent to easily pass through the mesh. Kogure et al. reported a technical success rate of 96% using the SIS method in a single session, which is similar to that in our study [19]. Therefore, advances in stent design can facilitate bilateral metal stent insertion in the future. In terms of patency, as there was no control group, it was impossible to evaluate its superiority. However, our result (median: 199 days) seems to be comparable to the findings of other studies involving bilateral stenting [14, 19–21].

There are several limitations in our study. First, it was a retrospective, non-comparative design and included a small sample size. However, malignant hilar obstruction is not a common disease. A sample size of more than 80 patients with malignant hilar obstruction would be sufficient to assess stent patency and predictive factors. A prospective, randomized trial will be needed in the future. Second, there were various types of cancers with different characteristics. However, most of the diseases in the enrolled sample were CCA. Moreover, as sepsis from biliary obstruction is important in most of the outcomes of patients with malignant hilar obstruction, this is not excluded in the assessment of stent patency. Third, as the interval from nasobiliary drainage to metal stent insertion was not standardized, each case of bilateral stent replacement was performed with different intervals. Additionally, chemotherapy and radiation therapy regimens were heterogeneous. Fourth, the effects of local therapies such as intra-ductal radiofrequency ablation (RFA) and photodynamic therapy were not analyzed. Recently, it was found that bile duct carcinoma was associated with an increased frequency of intra-ductal RFA [22]. An analysis of the effects of local therapies on stent patency and survival is needed in the future. Finally, our study included patients with only large cell-type metal stents, which might limit the generalizability of the results. Patency may vary depending on the plastic stent, type of metal stent, and method of bilateral stent replacement. Our study aimed to investigate factors associated with stent obstruction with only large cell-type stents. In a retrospective study by Lee et al., there was no significant difference in stent patency according to the cell size of the metal stents (small cell-sized vs. large cell-sized stents, 42.9% vs. 45.5%, p = 0.086) [23]. Further investigation is necessary to compare several parameters among various types of stents and various methods of stent replacement. Nevertheless, the strength of our study is that a

single stent model was used with the same technique; thus, the risk of bias associated with the stent type and technique could be reduced.

In conclusion, cholangitis was associated with shorter patency, and subsequent chemotherapy was associated with longer patency. Further studies are required to identify the optimal timing and indication for bilateral stent insertion to improve stent patency in patients with malignant hilar obstruction.

## Supporting information

**S1 File.**
(DOCX)

**S1 Data.**
(XLSX)

## Author Contributions

**Conceptualization:** Tae Jun Song, Seok Ho Dong.

**Formal analysis:** Hoonsub So, Tae Jun Song.

**Investigation:** Hoonsub So, Tae Jun Song, Seok Ho Dong.

**Methodology:** Sung Woo Ko, Jun Seong Hwang, Dongwook Oh, Do Hyun Park, Sang Soo Lee, Dong-Wan Seo, Sung Koo Lee, Myung-Hwan Kim.

**Supervision:** Sung Koo Lee, Myung-Hwan Kim.

**Writing – original draft:** Hoonsub So, Chi Hyuk Oh.

**Writing – review & editing:** Hoonsub So, Chi Hyuk Oh, Tae Jun Song.

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
