## [Decision Letter · Decision Letter 0]

13 Nov 2020

PONE-D-20-33082

Predictors of stent dysfunction in patients with bilateral metal stents for malignant hilar obstruction

PLOS ONE

Dear Dr. Tae Jun Song,

Thank you for submitting your manuscript to PLOS ONE. After careful consideration, we feel that it has merit but does not fully meet PLOS ONE’s publication criteria as it currently stands. Therefore, we invite you to submit a revised version of the manuscript that addresses the points raised during the review process.

Please submit your revised manuscript within 60 days. If you will need more time than this to complete your revisions, please reply to this message or contact the journal office at plosone@plos.org. Please include the following items when submitting your revised manuscript:

We look forward to receiving your revised manuscript.

Kind regards,

Gianfranco D. Alpini

Academic Editor

PLOS ONE

Journal Requirements:

2.) In the ethics statement in the manuscript and in the online submission form, please provide additional information about the patient records used in your retrospective study, including: a) whether all data were fully anonymized before you accessed them; and b) the date range (month and year) during which patients' medical records were accessed. If patients provided informed written consent to have data from their medical records used in research, please include this information.

3.)Thank you for stating the following financial disclosure:

 [The funders had no role in study design, data collection and analysis, decision to publish, or preparation of the manuscript.].

4.)  In your Data Availability statement, you have not specified where the minimal data set underlying the results described in your manuscript can be found. PLOS defines a study's minimal data set as the underlying data used to reach the conclusions drawn in the manuscript and any additional data required to replicate the reported study findings in their entirety. All PLOS journals require that the minimal data set be made fully available. For more information about our data policy, please see http://journals.plos.org/plosone/s/data-availability.

5.) PLOS requires an ORCID iD for the corresponding author in Editorial Manager on papers submitted after December 6th, 2016. Please ensure that you have an ORCID iD and that it is validated in Editorial Manager. To do this, go to ‘Update my Information’ (in the upper left-hand corner of the main menu), and click on the Fetch/Validate link next to the ORCID field. This will take you to the ORCID site and allow you to create a new iD or authenticate a pre-existing iD in Editorial Manager. Please see the following video for instructions on linking an ORCID iD to your Editorial Manager account: https://www.youtube.com/watch?v=_xcclfuvtxQ

Reviewers' comments:

Reviewer's Responses to Questions

**Comments to the Author**

1. Is the manuscript technically sound, and do the data support the conclusions?

Reviewer #1: Yes

Reviewer #2: Yes

2. Has the statistical analysis been performed appropriately and rigorously? 

Reviewer #1: Yes

Reviewer #2: Yes

3. Have the authors made all data underlying the findings in their manuscript fully available?

Reviewer #1: Yes

Reviewer #2: No

4. Is the manuscript presented in an intelligible fashion and written in standard English?

Reviewer #1: Yes

Reviewer #2: Yes

5. Review Comments to the Author

Reviewer #1: I read the manuscript So et al with great interest. The authors sought to verify the efficacy of double metal stents in a high risk patients with cholangiocarcinoma (CCA). IN fact, as noted also by authors this was done mainly for palliative reasons since there was a >50% death within less than a year. The median follow was less than a year.

I think study is interesting to show what this group expertise with double stenting. However, since it is a retrospective study, it would be very useful to add a group to compare plastic stents or PTC drains. I believe the group should have some historical patients with plastic stents / or PTC drains or common-bile duct single metal stents. One of these groups or all will increase the authors' thesis on the use of double metal stents.

Minor: common abbreviation for cholangiocarcinoma is CCA not CCC. please use CCA throughout the manuscript.

Reviewer #2: In the current manuscript, So et al. aimed to study the association of stent dysfunction and predictors related to stent dysfunction and overall survival in patients with malignant hilar obstruction. The authors retrospectively analyzed these associations in patients (n=87) who underwent bilateral metal stent placement by using large-cell-type stents at two academic teaching hospitals. Among them, it was found that stent dysfunction occurred in 42 patients (48.3%) and the median stent patency was 199 days. Besides, age, cholangitis before stent insertion, and subsequent chemotherapy were found to be associated with the cumulative risk of stent dysfunction. Overall, the current study has some clinical importance in guiding the application of bilateral metal stent placement in patients with malignant hilar obstruction. However, some concerns should be further clarified by the authors.

1. From the introduction, uncovered stents are more likely to be re-obstructed by tumor ingrowth. The current aimed to investigate factors associated with stent obstruction in a large-cell-type stent which is one type of uncovered stents. Based on this description, the reviewer expects to see the comparison of several parameters among large-cell-type and other types of stents. However, it is not present in the current study.

2. The study aimed to investigate factors associated with stent dysfunction in patients with malignant hilar obstruction. As suggested by the authors, the endoscopic placement of stents is currently a mainstay of its treatment in unresectable malignant hilar biliary obstruction. To improve the significance of this study, it is required to further analyze the difference between other therapies such as chemotherapies which are used for unresectable malignant hilar biliary obstruction and endoscopic placement of a large-cell-type stent.

6. PLOS authors have the option to publish the peer review history of their article (what does this mean?). If published, this will include your full peer review and any attached files.

Reviewer #1: No

Reviewer #2: No

---

## [Author Response · Author response to Decision Letter 0]

21 Feb 2021

Cover letter with point-by-point responses

Dear Editor-in-Chief and reviews

Thank you for reviewing our paper titled “Predictors of stent dysfunction in patients with bilateral metal stents for malignant hilar obstruction”. We appreciate the time you have taken to review our manuscript. We have made some corrections following your helpful comments, and we believe our paper has been much improved.

Therefore, we would like to resubmit this revised manuscript to “PLOS ONE” for publication. Point-by-point responses to the comments of the reviewers have also been prepared. All authors agreed to accept equal responsibility for the accuracy of the content of the paper. We hope the revised manuscript will better meet the requirements of “PLOS ONE” for publication and be finally accepted. 

The detailed response of the reviewer's comments will be explained in a separate file.

We thank you again for your thoughtful consideration.

Sincerely,

Tae Jun Song, on behalf of

Division of Gastroenterology

Asan Medical Center, University of Ulsan College of Medicine, Seoul, Korea

E-mail: drsong@amc.seoul.kr

---

## [Decision Letter · Decision Letter 1]

11 Mar 2021

Predictors of stent dysfunction in patients with bilateral metal stents for malignant hilar obstruction

PONE-D-20-33082R1

Dear Dr. Tae Jun Song,

We’re pleased to inform you that your manuscript has been judged scientifically suitable for publication and will be formally accepted for publication once it meets all outstanding technical requirements.

Kind regards,

Gianfranco D. Alpini

Academic Editor

PLOS ONE

Additional Editor Comments (optional):

Reviewers' comments:

Reviewer's Responses to Questions

**Comments to the Author**

1. If the authors have adequately addressed your comments raised in a previous round of review and you feel that this manuscript is now acceptable for publication, you may indicate that here to bypass the “Comments to the Author” section, enter your conflict of interest statement in the “Confidential to Editor” section, and submit your "Accept" recommendation.

Reviewer #2: All comments have been addressed

2. Is the manuscript technically sound, and do the data support the conclusions?

Reviewer #2: Yes

3. Has the statistical analysis been performed appropriately and rigorously? 

Reviewer #2: Yes

4. Have the authors made all data underlying the findings in their manuscript fully available?

Reviewer #2: Yes

5. Is the manuscript presented in an intelligible fashion and written in standard English?

Reviewer #2: Yes

6. Review Comments to the Author

Reviewer #2: Thanks for your responses. All my concerns have been addressed. The current manuscript is acceptable.

7. PLOS authors have the option to publish the peer review history of their article (what does this mean?). If published, this will include your full peer review and any attached files.

Reviewer #2: No

---

## [Editor Report · Acceptance letter]

18 Mar 2021

PONE-D-20-33082R1 

Predictors of stent dysfunction in patients with bilateral metal stents for malignant hilar obstruction 

Dear Dr. Song:

I'm pleased to inform you that your manuscript has been deemed suitable for publication in PLOS ONE. Congratulations! Your manuscript is now with our production department. 

Kind regards, 

on behalf of

Dr. Gianfranco D. Alpini 

Academic Editor

PLOS ONE